# Effect of Endometrial Sampling Procedures on Subsequent Pregnancy Rate of Cattle

**DOI:** 10.3390/ani11061683

**Published:** 2021-06-05

**Authors:** Orlando Ramirez-Garzon, Ricardo Soares Magalhaes, Nana Satake, Jonathan Hill, Claudia Jimenez, Michael K. Holland, Michael McGowan

**Affiliations:** 1School of Veterinary Science, The University of Queensland, Gatton, QLD 4343, Australia; n.satake@uq.edu.au (N.S.); jonathan.hill@uq.edu.au (J.H.); mike.holland@uq.edu.au (M.K.H.); m.mcgowan@uq.edu.au (M.M.); 2Facultad de Medicina Veterinaria, Universidad Nacional, Bogota 111321, Colombia; cjimeneze@unal.edu.co; 3Child Health Research Centre, The University of Queensland, South Brisbane, QLD 4141, Australia; r.magalhaes@uq.edu.au

**Keywords:** cattle, cotton swab, endometrial biopsy, cytobrush, cytotape, uterine lavage

## Abstract

**Simple Summary:**

An important concern about conducting endometrial sampling procedures prior to breeding is the potential adverse effect on subsequent fertility of sampled cattle. Traumatic injury to the cervix and uterus and risk of introduction of infection caused by these sampling procedures may adversely affect the fertility of cattle sampled prior to breeding. To investigate this, a systematic review of publications assessing the impact of endometrial sampling on subsequent pregnancy rates was conducted. Analysis within the studies selected demonstrated that pregnancy rates were similar between sampled and non-sampled animals when procedures were performed before or around the time of breeding. Unfortunately, it was not possible to compare each sampling procedure between studies due to variation in sampling conditions (i.e., type of breed, age, reproductive status, and the sampling to breeding interval). We concluded that conducting these endometrial sampling procedures does not adversely affect subsequent fertility; therefore they could be used to more accurately identify cattle with a normal endometrium prior to conducting procedures such as embryo transfer. However, further studies with a much larger number of cattle are needed to verify the effects of endometrial sampling on pregnancy rates.

**Abstract:**

Endometrial infections are a common cause of reproductive loss in cattle. Accurate diagnosis is important to reduce the economic losses caused by endometritis. A range of sampling procedures have been developed which enable collection of endometrial tissue or luminal cells or uterine fluid. However, as these are all invasive procedures, there is a risk that sampling around the time of breeding may adversely affect subsequent pregnancy rate. This systematic review compared the pregnancy rates (PR) of cattle which underwent uterine lavage (UL), cotton swab (CS), cytobrush (CB), cytotape (CT), or endometrial biopsy (EB) sampling procedures with those that were not sampled. Using the Preferred Reporting Items for Systematic Reviews and Meta-Analysis (PRISMA) protocol, relevant databases, including Pubmed, Web of Science, CAB Abstracts, VetMed Resource–Ruminants, and Scopus, were searched. The outcome measured was the pregnancy rate after the collection of endometrial sample(s). Seven studies, involving a total of 3693 cows, fulfilled the inclusion criteria for the systematic review and allowed the comparison of PR between sampled (*n* = 1254) and non-sampled cows (*n* = 2409). The results of the systematic review showed that endometrial sampling procedures can be performed before breeding or shortly after insemination without adversely affecting pregnancy rates in cattle. However, further studies are needed to validate this information.

## 1. Introduction

High reproductive performance in production animals such as beef and dairy cattle is vital for achieving optimal per capita return. Endometritis is a common cause of reproductive failure, especially in dairy cattle, causing increases in both calving to conception interval and culling rates [1,2]. Therefore, detection of endometritis in individual cows, before breeding or embryo transfer (ET), is critical.

The diagnosis of endometritis often relies on the detection of purulent or mucopurulent vulvar or cervical discharge, or palpation of enlarged, sometimes fluctuant uterine horns which lack tone. These clinical signs are usually detected by vaginoscopy [3,4] transrectal palpation [5,6], and/or ultrasound [7,8,9]. These quick, low-cost diagnostic methods are commonly employed in routine herd health postpartum examinations. However, these methods underestimate the prevalence of subclinical endometritis [10]. Histological changes to the endometrium, such as increased presence of inflammatory cells in subclinical endometritis, can only be detected by cytology or histopathology [11]. Hence, more invasive sample collection methods such as uterine lavage (UL), intrauterine cotton swab (CS), cytobrush (CB), or cytotape (CT) sampling, and endometrial biopsy (EB) are required to confirm the diagnosis. These techniques enable the collection of epithelial and inflammatory cells (CS, UL, CB, and CT), luminal secretions (UL), and endometrial tissue (EB) that allow the inspection of deeper physiological and cellular responses not yet identifiable by routine clinical examinations. The samples obtained can be subjected to cytological examination [12,13], bacteriological culture [14], histopathological examination [15], protein analysis [16], and gene expression analysis [17] to diagnose the status of the endometrial environment.

Collectively, these methods involve a transvaginal device being inserted through the cervix (using per rectal manipulation) into the uterine body or uterine horns to collect the sample required (Figure 1). Briefly, for UL, a sterile catheter is introduced into the uterine horn and 20–50 mL of sterile 0.9% sodium chloride solution is infused, and then after per rectal massage of the uterine horns, the saline is aspirated [7,18]. A sterile disposable cotton swab is normally attached to the tip of a stylet of a bovine artificial insemination (AI) gun, which is then enclosed by an outer protective plastic sheath. Once in the uterine lumen, the swab is advanced through the plastic sheath and then moved backwards and forwards against the endometrium. Before removal, the swab is pulled back into the insemination gun to avoid cervical and vaginal contamination [6,19]. The CB [20,21] (Cytology Brush; Minitube GmbH, Germany) is a disposable semi-rigid device, protected by a catheter that is inserted into the uterus. Within the uterus, the brush is pushed forward and rotated clockwise along the uterine wall. The CT device consists of a section of rolled paper tape attached to the tip of an AI gun stylet with the AI gun protected by a plastic sheath. Once in the uterine lumen, the rolled tape is advanced through the plastic sheath and rotated against the uterine walls [13]. For EB, the device is guided into the uterine horn, the forceps jaws are then opened and a section of the uterine wall gently pushed into the jaw and closed [15,22]. The size of the biopsy varies according to the device’s jaw size. Recently, a new sampling device which allows the collection of endometrial cells, tissue, and uterine secretions after a single passage through the cervix has been developed [23].

The degree of endometrial injury and trauma varies with the method of sampling from likely to negligible for UL, CS, CB, and CT, to potentially moderate damage when performing EB. EB involves collection of a full thickness section of the endometrium, and in some cases a portion of the underlying myometrium (depth of tissue varies from 0.4 to 1 cm) [15,24]. In mares [25] and women [26], endometrial sampling is a routine procedure which does not apparently adversely affect the likelihood of the sampled female becoming pregnant. However, the impact of these procedures on the subsequent reproductive performance of cattle is still unclear. The objective of this study was to systematically review and summarize existing evidence related to the impact of these endometrial sampling procedures on the likelihood of sampled cattle becoming pregnant.

## 2. Materials and Methods

A systematic review was conducted of studies investigating the association between pregnancy rates after collection of endometrial samples from cattle by different methods (UL, CS, CB, CT, and EB), using the Preferred Reporting Items for Systematic Reviews and Meta-Analysis (PRISMA) protocol. Online searches of literature databases were carried out in Pubmed, Web of Science, CAB Abstracts, VetMed Resource–Ruminants, and Scopus with no date limitations. A combination of the following search words was used to generate a subset of citations: cattle OR cow OR bovine OR buffalo OR bos AND uterus OR endometrial AND cytology OR cytobrush OR “cotton-swab” OR “uterine lavage” OR “uterine aspiration” OR biopsy AND fertility OR “pregnancy rate” OR “reproductive performance.” Primary papers written in English, Portuguese, and Spanish were accepted from peer-reviewed journals with no date restrictions. The latest date for search was 20 February 2021.

To avoid confusion due to the large variability in terminology, an invasive endometrial sampling refers to any procedure that retrieves a sample from the uterine horns. Thus, transrectal palpation and ultrasonographic examination of the uterus, and prevaginal examination or sample collection were not considered in this review.

Eligibility criteria included (1) all types of studies, including observational, experimental, and descriptive; (2) all study settings and countries; (3) studies where an endometrial sample was collected from cattle regardless of whether they had clinical signs of endometritis or not; and (4) studies reporting the pregnancy rate after the endometrial sampling was performed. The full text was then examined and retained if (1) endometrial samples were collected either by UL, CS, CB, CT, or EB around the time of breeding or embryo transfer; (2) the pregnancy rates reported were associated only with the sampling procedure itself; and (3) the pregnancy rates of sampled animals were compared to non-sampled animals.

Duplicate citations were excluded. Titles and abstracts from non-cattle species (goat, sheep, horse, and human) studies using in vitro procedures, post-mortem material or intrauterine infusions were also excluded. Citations were also excluded if the information was published in reviews, book chapters or conference proceedings, although their reference lists were examined for additional studies not identified by the primary search strategy. Abstracts were excluded if (1) uterine sampling procedures were used to establish the prevalence and/or threshold values for the diagnosis of clinical or subclinical endometritis; and (2) the reproductive performance was not assessed. Pregnancy rate (%; PR) was the primary outcome measure.

All articles selected from the electronic searches and data extraction were assessed by two authors (ORG and RSM). The final decision on the studies to include in the analysis was ORG’s. The following information from each study was tabulated: first author, year of publication, number of subjects (sampled and control), method of sampling, number of samplings, and pregnancy rate for sampled and control cattle.

## 3. Results

The studies were selected and reported according to the PRISMA 2009 guidelines (Figure 2). A total of 729 studies were identified, and 235 duplicate references were discarded. After reading the title and applying the exclusion criteria, 223 studies were excluded. A total of 271 citations were screened and 215 were excluded after reading the abstracts. Fifty-six publications were retrieved for a full text appraisal, with 49 subsequently excluded because they did not meet the predefined inclusion criteria (Table 1).

From the excluded manuscripts, 6 did not have control groups, 1 did not sample the endometrium, 3 papers used the same results in different publications, 3 papers were retrospective studies, 7 used the sampling methods to establish threshold values for diagnosis of endometritis, 12 did not report the pregnancy rates after sampling, and 16 investigated the impact of endometritis on pregnancy rates.

Seven studies including 3693 animals (sampled = 1254; not sampled = 2409) met the inclusion criteria. The characteristics of the selected studies are summarized in Table 2. Four of these were prospective cohort studies [18,20,27,28]. Two studies were randomized controlled studies [29,30] and one a case control study [31]. In the prospective cohort studies, the experiments compared the effect of CB [20], UL [18], or UL and EB [27] with no endometrial sampling. Within the randomized studies, one study compared EB [29] and the other compared UL [30] with no endometrial sampling. In the case control study [31], the effect of EB on the first AI pregnancy rates was compared with non-biopsied cows at 150 DIM (days in milk).

### 3.1. Uterine Lavage Studies

Cheong et al. [18] performed a prospective cohort study comparing the effect of UL to collect endometrial cells (*n* = 705) with no endometrial sampling (*n* = 1992) studying the reproductive performance of healthy Holstein cows. The selection criteria included primiparous and multiparous cows within 40–60 d postpartum, not inseminated without vaginal discharge or systemic illness. The reproductive performance was assessed during a 210-day period after endometrial sampling. The mean interval from sampling to first service was 19.4 days. In primiparous cows, the PR to first service was lower in sampled cows compared to cows which were not sampled (31.2% vs. 36.5% OR for pregnancy = 1.03; 95% C.I. 0.80–1.33; *p* = 0.82), whereas in multiparous cows, PR was similar in both groups (29.1% and 28.1% sampled and non-sampled cows, respectively).

In a randomized controlled study, Thome et al. [30] evaluated the effect of collecting endometrial cells by UL in postpartum Nellore cows (50–70 days postpartum). In 35 cows, the UL was performed 4 h after timed artificial insemination, while 93 were not sampled. No significant differences in PR were found between sampled and non-sampled groups (54.2% vs. 56.7%, respectively, *p* > 0.05).

Martins et al. [28] in a prospective cohort study compared the PR after ET on day 7.5 in non-lactating, cycling Nellore cows after having UL on days 1, 4, and 7 (*n* = 46; day 0; estrous detection) with control cows (*n* = 16). Control and day 1 cows (62.5% and 60%, respectively) had higher PR than cows sampled on day 4 and day 7 (29.4% and 37.5%, respectively) (*p* = 0.06).

### 3.2. Cytobrush Studies

In a prospective cohort study, Kaufman et al. [19] evaluated the effect of CB sampling the endometrium 4 h after artificial insemination on pregnancy rate to first service in cows calved at least 65 days. PR was similar for sampled and non-sampled cows (43.3% vs. 41.7%, *p* > 0.05), although significantly higher in primiparous than multiparous cows (54.3 vs. 38.5%, *p* < 0.05).

### 3.3. Endometrial Biopsy Studies

In a case control study, Goshen et al. [31] randomly selected 54 Holstein cows calved approximately 67 days to undergo EB; 157 control cows were paired with sampled cows. The effect of the biopsy on PR to first artificial insemination was calculated using binary logistic regression. The interval from biopsy to first AI was 40.5 days (range 5–111 days). The PR and days from calving to conception in biopsied cows (44.4%; 147.3 days) did not differ significantly from those in control cows (38.9%, 150.8 days).

Etherington et al. [29] conducted a randomized controlled trial on 130 postpartum dairy cows and evaluated the effect of postpartum EB between days 26 and 40 postpartum on PR to first AI and calving to conception interval. EB increased the interval from calving to first service (89 days biopsied cows versus 81.5 days for control cows; *p* = 0.07). However, the PR to first AI for biopsied cows (*n* = 92; 37%) was not significantly different from non-biopsied cows (*n* = 69; 39%).

In a prospective cohort study, Pugliesi et al. [27] evaluated the effect of UL (*n* = 35) and EB (*n* = 38) from the horn contralateral to the corpus luteum (CL) on day 6 after timed artificial insemination on pregnancy rates on days 30 and 60 in multiparous *Bos indicus* cows. After the procedure, all cows received a non-steroidal anti-inflammatory treatment (flunixin meglumine, 1.1 mg/kg bw, IM) and an antibiotic (penicillin–streptomycin 6,000,000 IU). The PR were similar (*p* > 0.1) for UL, EB, and non-sampled cows on day 30 (28.6, 31.6 and 40.5%, respectively), but the pregnancy rates decreased significantly in UL cows (*p* < 0.004) compared to control and EB cows on fay 60 (17.1%, 26.3%, and 40.5%, respectively).

## 4. Discussion

The aim of this review was to determine the likely impact of endometrial sampling procedures on the subsequent pregnancy rates of treated cattle. The results indicate that PR were similar for sampled and non-sampled animals using CB, UL, and EB if the procedure was performed before breeding or a few hours after insemination. However, it was affected if it was performed during early diestrus. The heterogeneity of the selected studies in terms of type of cattle used (dairy and beef), physiological stage during sampling (early postpartum [29] or late postpartum > 45 days [31], and the interval from sampling to breeding (i.e., 4 h [20,30] or 20–40 days [31] did not allow comparison between different methods to determine whether one method affects the PR more than any other. The CT [59,60] studies were not assessed as pregnancy rates from untreated controls were not included in the study.

These results suggest that perturbations caused after endometrial sampling might induce acute changes in the endometrial environment, but the ability to support embryo development and maintain a pregnancy is recovered. As an indirect indicator of uterine response to artificial insemination [71], changes in uterine blood flow have been measured using color Doppler transrectal ultrasonography. An increase in uterine blood flow was observed within 4 h of the procedure, which returned to baseline by 24 h, indicating that these procedures may induce a short acute inflammatory response. Although previous reports indicate that performing UL induces endometrial irritation caused either by the fluid [72] or by the device [73], studies in mares [74,75] and women [76] have demonstrated that UL did not induce significant morphological changes to the endometrial tissue [77]. Just before artificial insemination, Pascottini et al. collected endometrial cells using CT in nulliparous heifers [59] and multiparous cows [60], and then observed pregnancy rates of 62% and 43%, respectively, which are similar to pregnancy rates reported in non-sampled dairy cows [78]. Similarly, Cheong et al. [18] and Thome et al. [30] performed UL 4 h after insemination without affecting pregnancy rates which is consistent with results in other species such as horses where fertility is not reduced by post-breeding UL [79]. In ET studies, CB sampling one cycle before transfer (74) or collecting UL on day 1 postoestrus during the ongoing cycle [28] did not affect the pregnancy rates after ET. Therefore, it seems likely that recovering endometrial fluid or cells did not adversely affect fertilization and early embryo development but the time of sampling should be considered to allow the endometrial environment to recover after it is disturbed so as to not affect pregnancy outcome.

Sampling the endometrium close to the time of embryo arrival in the uterine horn (days 4–7) adversely affects embryo survival. In this review, one study [27] assessed the PR after performing UL or EB from the horn contralateral to the CL six days after insemination. Although PR were not significantly different on day 30 between sampled and non-sampled cows, the PR on day 60 were significantly lower in cows that had UL performed 6 days after insemination. In another study [28], the PR were reduced by about 50% by performing UL on day 4 or day 7 before ET (day 7.5 postoestrus). This increase in embryo and early foetal mortality can be attributed to either the early removal of unknown histotrophic factors needed for conceptus survival or by the endometrial inflammatory reaction and the subsequent influx of plasma proteins into the uterine lumen caused by the procedure [28]. The histotroph contains proteins, amino acids, and lipids essential for the support of early embryo development [80], and any induced change in the histotroph as a result of endometrial sampling may subsequently adversely affect early placental development [81].

One might speculate that after using more invasive procedures such as EB, the trauma and subsequent inflammatory and wound healing reaction will adversely affect embryo development and survival. However, the findings from the studies in this review showed that PR after EB were not affected by the type of cattle (beef or dairy), uterine and ovarian stage (uterine involution/anoestrus), parity (heifers or multiparous cows), number of interventions (single or multiple procedures), and the type of device used. Etherington et al. [29] found that cows that underwent EB from both horns on day 26 and/or day 40 had a longer calving to conception interval compared to non-biopsied cows (135 vs. 115 days, respectively, *p* = 0.03), and prolonged interval from calving to first service compared to non-biopsied cows (89 vs. 81 days, respectively, *p* = 0.07). However, they did not find significant variation in the pregnancy rates between biopsied and non-biopsied cows (37% vs. 39%, respectively). In a larger study of high-yield Holstein cows (*n* = 54), Goshen et al. [31] evaluated the effect of biopsying high-yield milking cows after uterine involution (between 44 and 104 days postpartum) on pregnancy rates. They did not find significant differences in PR between biopsied and control cows (44.4 vs. 38.9, respectively, *p* = 0.146,). These results suggest that performing biopsies before the completion of uterine involution may slow down the complete recovery of the endometrium postpartum, but if the biopsy is performed after involution, it apparently does not affect the conception rates.

In other studies that were not considered in the systematic review for not fulfilling the selection criteria (Table 1), pregnancy rates have been assessed after EB. Chapwanya et al. [15] evaluated the effect of three consecutive endometrial biopsies in the same uterine horn at days 15, 30, and 60 on pregnancy rates in postpartum cows (*n* = 13). They reported pregnancy rates of 77% performing the first AI 30 days after the last biopsy. Similarly, Rhoads et al. [24] collected three EB per cow (*n* = 33) from both horns at different times (3 days before oestrus, during oestrus, and 4 days after insemination) and reported PR in biopsied cows of 52%, which is considered within the normal range of PR in dairy postpartum cows. Similarly, Meikle et al. [82] and Katagiri et al. [83] performed four and six EB, respectively, within the oestrus cycle in the same animal and found that the majority of biopsied animals became pregnant within the first two detected oestrus after sampling (*n* = 5/7 PR 71% [82] and *n* = 14/25 PR 56% [83]), although the interval from sampling to onset of oestrus or the number of days to return to cyclicity was not mentioned. Collectively, these reports suggest that biopsy is a safe procedure which does not have a deleterious effect on fertility and PR.

## 5. Conclusions

The results of this systematic review show that invasive methods of endometrial sampling can be performed before breeding or within the first day postoestrus without affecting pregnancy rates in cattle with a healthy endometrium. However, caution must be taken since comparison between studies were not possible, and further studies with much larger numbers of cattle are needed to verify the effect of endometrial sampling on pregnancy rates.

## Figures and Tables

**Figure 1 animals-11-01683-f001:**
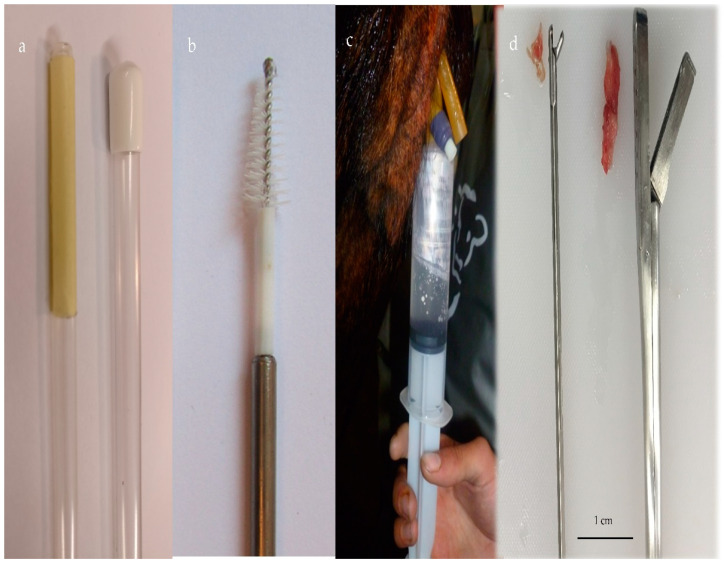
(**a**) Cytotape (Credit: Osvaldo Bogado Pascottini), (**b**) cytobrush, (**c**) uterine lavage with a saline solution using a Foley catheter, and (**d**) endometrial tissue with two biopsy devices.

**Figure 2 animals-11-01683-f002:**
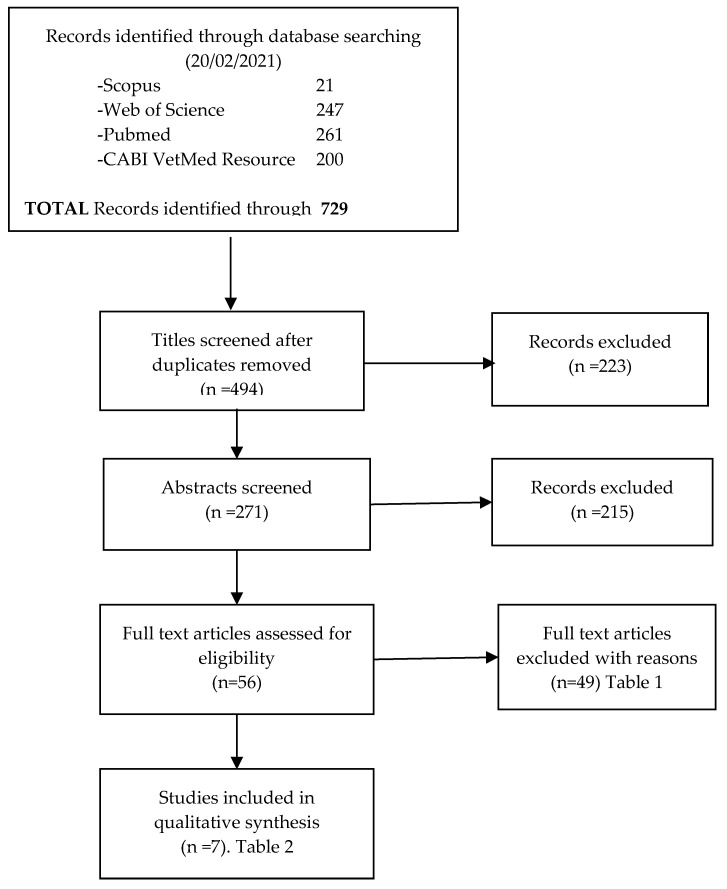
Selection process of papers for systematic review.

**Table 1 animals-11-01683-t001:** Characteristics of studies excluded.

Study	Type of Intervention	Criteria
Bacha and Regassa, 2010 [32]	UL	Pregnancy rate only from cows with endometritis diagnosed by cytology
Baranski et al., (a) 2012 [33] and (b) 2013 [34]	CB	(a) Established threshold values for diagnosis of subclinical endometritis. (b) Determined the impact of cytological endometritis on pregnancy rates
Barlund et al., 2008 [7]	CB, UL	Compared thresholds of different techniques for diagnosis of endometritis
Barrio et al., 2015 [35]	CB	Diagnosed subclinical endometritis by Cytobrush and determined the impact on reproductive performance
Bicalho et al., 2016 [36]	EB	Pregnancy rate not mentioned
Binelli et al., 2015 [37]	EB	Transcriptome uterine analysis was retrospectively performed in pregnant and non-pregnant cows
Bolzenius et al., 2016 [38]	EB	Pregnancy rate not compared with non-biopsied cows
Brodzki et al., (a) 2015 [39], (b) 2014 [40],	(a) UL, CB, (b) CB	(a) Pregnancy rate not described. (b) Pregnancy rates were related to presence/absence of subclinical endometritis
Carneiro et al., (a) 2013 [41], (b) 2014 [42]	(a)UL, (b) CB	(a) Evaluate the cytological endometritis diagnosed by uterine lavage in reproductive performance. (b)Evaluate reproductive performance in cows diagnosed with subclinical endometritis
Chapwanya et al., 2009 [43]; 2010 [15]; 2012 [44]	EB	No control group. Same data in three studies
Cheong et al., (a) 2011 [45], (b) 2012 [46]	UL	(a) Determine risk factors for subclinical endometritis and its effect on reproductive performance. (b) Determine the use of leukocyte esterase strip as indicator of endometritis
Couto et al., 2013 [47]	CB, UL	Used uterine fluid to measure leukocyte esterase activity to diagnose endometritis
de Biase et al., 2018 [48]	EB	Diagnosis of infectious agents using EB but pregnancy rate was not reported
de Boer et al., 2015 [49]	CB	Assessed % PMN cells and PR given on cows with endometritis
De Sa et al., 2017 [50]	EB	Assess transcriptome analysis of endometrium on day 6 from pregnant and non-pregnant cows
Denis-Robichaud, 2015 [51]	CB	Used the leukocyte esterase test for diagnosis of endometrial health and as predictor of pregnancy status in cows with endometritis
Gabai et al., 2019 [52]	CB and UL	Pregnancy rate not described
Kasimanickam et al., (a) 2004 [12], (b) 2006 [53]	(a) CB (b) None	(a) Validated cytology for diagnosis of endometritis (b) They did not use any endometrial sampling procedure
Katagiri et al., 2006 [54]	EB	Pregnancy rate given but no control group
Lopez et al., 2012 [2]	CB	Pregnancy rate associated with endometrial health postpartum
Machado et al., 2012 [55]	UL	Pregnancy rate associated with endometrial health
Madoz et al., 2013 [56]	CB	Used endometrial cells to establish cut off values for endometritis in grazing cows
McDougall et al., 2011 [57]	CB	Pregnancy rate correlated with % PMN
Nehru et al., 2019 [58]	CB and UL	Pregnancy rate reported but no control group
Pascottini et al., 2015 [13]	CB, CT	Pregnancy rate not described
Pascottini et al., 2016 [59]	CT	Pregnancy rate not reported in non-sampled heifers
Pascottini et al., 2017 [60]	CT	Pregnancy rate not reported in non-sampled cows
Plontzke et al., 2010 [9]	CB	Pregnancy rates associated with % PMN in subclinical endometritis
Prunner et al., 2014 [61]	CB	Pregnancy rate determined by postpartum uterine health
Rhoads et al., 2008 [24]	EB	Pregnancy rate not reported
Ricci et al., 2015 [62]	UL	Pregnancy rate associated with %PMN in subclinical endometritis
Salasel et al., 2010 [63]	UL	Pregnancy rate associated with subclinical endometritis and risk factors
Santos et al., 2009 [64]	UL	Pregnancy rates associated with % PMN in *Bos indicus* cows
Scolari et al., 2017 [65]	EB	Assess transcriptome analysis of endometrium on day 6 from pregnant and non-pregnant cows
Senosy et al., 2012; [66]	CB	Fertility rates based on uterine health diagnosed postpartum
Sens and Heuwieser, 2013 [67]	CB	No control group
Studer et al., 1978 [6]	EB	Pregnancy rate is not mentioned after procedure
Van Schyndel et al., 2019 [68]	CB, UL	Compared cytologies for diagnosis of subclinical endometritis, but pregnancy rate is not described
Werner et al., 2012 [69]	CB	Pregnancy rate is not mentioned after procedure
Westerman et al., 2010 [3]	CB	Pregnancy rate is not mentioned after procedure
Zaayer et al., 1986 [70]	EB	Pregnancy rate is not mentioned after procedure

UL: uterine lavage; CB: cytobrush; EB: endometrial biopsy; CT: cytotape; PMN: polymorphonuclear cells; PR: pregnancy rate; SCE: subclinical endometritis.

**Table 2 animals-11-01683-t002:** Pregnancy rates after endometrial sampling in cattle.

Author	Study Design	Sampled (*n*)	Control (*n*)	Intervention (#)	Interval Procedure-Breeding	PR Sampled (%)	PR Control (%)
Etheringthon et al. [29])	RCS	92	69	EB	NM	37	39
				(4)			
Kauffman et al. [20]	CS	201	103	CB	4 h	43.3	41.7
				(1)			
Cheong et al. [18]	CS	705	1992	UL (1)	19.4 d (SEM = 0.4)	PRI 31.2 MUL 29.1	36.5 28.1
Goshen et al. [31]	CC	44	157	EB (1)	40.5 d (5–111)	44.4	38.9
Pugliesi et al. [27]				EB (1)		EB 31.6; UL 28.6 **	40.5 **
	CS	73	37		6 d *		
				UL (1)		EB 26.3; UL 17.1 ***	40.5 ***
Thome et al. [30]	RCS	93	35	UL (1)	4 h after AI	57 ± 5.1	54 ± 8.4
Martins et al. [28])	CS	46	16	UL (1)	6 d, 3 d and 0.5 d ***	Day 1 (60) Day 4 (29.4) Day 7 (37.5)	62.5

RCS: randomized controlled study; CS: cohort study; CC: case control study; UL: uterine lavage; EB: endometrial biopsy; CB: cytobrush; AI artificial insemination;* time of breeding after intervention; ** 30 days of pregnancy assessment after breeding; *** 60 days of pregnancy assessment after breeding *** interval from UL to embryo transfer (ET)(day 7.5 postoestrus).

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
