# Peer review of "Effect of Endometrial Sampling Procedures on Subsequent Pregnancy Rate of Cattle"

_animals, 2021, doi:10.3390/ani11061683_

Round 1

Reviewer 1 Report

The authors acknowledged most of the points but I still have some critical remarks, which must be included in the discussion.

1) Point 1: L22-24: What does this statement mean for daily practice? Do the authors suggest not breeding cows with signs of inflammation?

Response 1: The statement was rephrased to make it clearer. The decision whether to breed an animal should be made following the completion of a reproductive tract assessment. This decision should not be based solely on uterine health. The statement refers to the importance of incorporating these sampling procedures into daily practice. This allows for detection of animals with uterine pathologies that are not diagnosed through conventional rectal palpation or scanning. In having the awareness that sampling and sample processing is time demanding, these procedures could be implemented in a cost-effective manner as routine diagnostic tests on cows or farms with fertility problems to improve pregnancy rates

1.1. do not agree with the general statement that the implementation of CB, CT, UL into daily practice would improve pregnancy rates. Too critical insemination management leads to decreased submission rates, which would in turn affect the pregnancy rate. The authors should critical discuss the consequences of the implementation of methods for the diagnosis of subclinical endometritis in daily practice for the breeding decision.

2) Point 2: L53-67: Authors write that cytobrush (CB), cytotape (CT), uterine lavage (UL) and endometrial biopsy (EB) are more accurate for the diagnosis of pathological conditions in the uterus (inflammation) than conventional methods used in practice, i.e., the evaluation if vulvar or cervical discharge or accumulation of uterine fluid. I disagree with this statement because for daily practice, it is primary relevant that a diagnosis is followed by a consequence (decision on treatment/non-treatment). Since for subclinical endometritis (SE), defined as endometrial inflammation without clinical symptoms, no evidence based treatment advice can be given, the diagnosis is not useful in practice because no consequence is followed. In addition, the decision not to breed an animal because of SE has be regarded with caution because of the poor accuracy to predict non-pregnancy in those animals. In this context, literature associating different signs of inflammation in histological samples taken by biopsy with fertility outcome is lacking. For research, however, results summarized in this review are highly relevant when accociating uterine findings after CT, CB, UL, EB sampling with the fertility outcome. Here, any effect of uterine sampling on fertility must be excluded. Therefore, authors should emphasize the importance of their review for research rather than for practice.

Response 2: The diagnosis and treatment of subclinical endometritis (SCE) is a consistent challenge and under discussion. Currently, there is much ongoing research to improve the diagnosis of uterine diseases through identification of endometrial biomarkers and proteins, which requires the collection of endometrial samples using the techniques described in the manuscript.

Considering SCE in daily practice, at the time of postpartum assessment, the threshold levels of PMN cells and the subsequent interpretation using the gold standard (reproductive outcome) is also controversial. In the literature, there is a high variation in the prevalence of

SCE (10-50%; Kasimanickam et al., 2004, Barlund et al., 2008, Pascotini et al., 2017) which mainly depends on the time of postpartum assessment and the cutting-off point for inflammatory cells (1-18%; Plontzke et al., 2010, de Boer et al., 2014). It has been demonstrated that SCE has a detrimental effect on reproductive performance (Wagener et al., 2017, Pascotini et al., 2017) because of the impact on survival and quality of the embryo (Gilbert et al., 2005). However, some authors did not find any consequential effects of SCE on reproductive performance (Plontzke et al., 2010, Santos et al., 2013). Because it is a variable that depends on many other factors (rather than uterine health), discrepancies in the results may be associated with the accuracy of using reproductive outcomes as a reference (gold standard) .

As the reviewer suggests, uterine sampling is currently the most taken for research purposes. In veterinary practice, a very limited number of farms are routinely using these techniques to diagnose uterine health. The extra labor, the cost, and time required to process the samples are the main reasons for not routinely evaluate uterine health on commercial farms. On the other hand this approach could be used in specific cases were clinicians could recommend further diagnostic tests

2.2. I agree that the reproductive outcome as gold standard is controversial. This is exactly the reason, why it is difficult to depend a single breeding decision on a threshold of PMN or histological finding. For sure, the diagnosis of subclinical endometritis could be implemented as a monitoring tool during herd health visits on farms with fertility problems. The prevalence of subclinical endometritis in a subgroup of cows can be used to identify and discuss risk factors for subfertility on the particular farm. Again, I strongly recommend to discuss the implementation of SE diagnosis into daily practice more critically.

Minor remarks

L2-3 The title is misleading since the review includes studies where samples were taken before or around the time of breeding.

L56-57 The methods do not underestimate the prevalence of subclinical endometritis, they cannot detect this disease, so it remains undiagnosed.

Fig. 1 the pictures b and c are still poor (incorrectly scaled).

Reviewer 2 Report

Dear authors,

After again critically reading the resubmitted manuscript including your improvements based on the reviewer comments again I may concluded the manuscript has been improved significantly and the provided reply comments for improvement have been satisfactory accepted and/or implemented. This means as a conclusion, I do not have any further (critical)  added comments on this resubmitted version in its current version.

Regards Reviewer-3

Reviewer 3 Report

The paper "Do endometrial sampling procedures conducted prior to breeding adversely affect the pregnancy rate of cattle" is a systematic review to assess the impact of several endometrial sampling procedures on preganncy rate in cattle. This reviewer has several concerns about this study. As the authors specified at the end of the abstract "Although comparisons between studies were not possible..:". Thus, it is difficult to accept the conclusion that "endometrial sampling procedures can be performed before breeding or shortly after insemination without adversely affecting pregnancy rates in cattle"!!! This reviewer consider that the information that can be received by the readers would be misleading  and not correct. For thsi reason the paper can not be acceptable.

Reviewer 4 Report

The manuscript needs to perform some English minor revision.

Round 2

Reviewer 3 Report

The paper can be acceptable for publication

This manuscript is a resubmission of an earlier submission. The following is a list of the peer review reports and author responses from that submission.

Round 1

Reviewer 1 Report

In this systematic review, authors summarize studies on the effect of uterine sampling on subsequent fertility. The topic is of interest to the readers of the journal Animals. The review is clearly written and well structured. Please find below some general criticisms on the rationale of using uterine sampling in daily practice.

L22-24: What does this statement mean for daily practice? Do the authors suggest not breeding cows with signs of inflammation?

L53-67: Authors write that cytobrush (CB), cytotape (CT), uterine lavage (UL) and endometrial biopsy (EB) are more accurate for the diagnosis of pathological conditions in the uterus (inflammation) than conventional methods used in practice, i.e., the evaluation if vulvar or cervical discharge or accumulation of uterine fluid. I disagree with this statement because for daily practice, it is primary relevant that a diagnosis is followed by a consequence (decision on treatment/non-treatment). Since for subclinical endometritis (SE), defined as endometrial inflammation without clinical symptoms, no evidence based treatment advice can be given, the diagnosis is not useful in practice because no consequence is followed. In addition, the decision not to breed an animal because of SE has be regarded with caution because of the poor accuracy to predict non-pregnancy in those animals. In this context, literature associating different signs of inflammation in histological samples taken by biopsy with fertility outcome is lacking. For research, however, results summarized in this review are highly relevant when accociating uterine findings after CT, CB, UL, EB sampling with the fertility outcome. Here, any effect of uterine sampling on fertility must be excluded. Therefore, authors should emphasize the importance of their review for research rather than for practice.

L98-99: Studies on the effect of EB on fertility are existing, so the sentence has to be reformulated.

L84: The quality of some pictures is poor

L142: The quality of some pictures is poor

L230: Please indicate the method of endometrial sampling

Author Response

Dear reviewer, 

Thank you very much for your time and suggestions to improve the manuscript. Please find attached the answers to your comments.

Best Regards, 

Orlando Ramirez-Garzon

DVM, MSc, PhD

School of Veterinary Science,

The University of Queensland, Gatton campus, 4343,

Australia

Reviewer 2 Report

General (major) comments

- The major concern about this review was the exclusion of a large number of papers just by reading the titles (first step – Systematic review), and not after reading the title and abstract together. I understand that one step has been skipped.  At least the exclusion and inclusion criteria should be made clearer in the methodology section.

- It is necessary to insert the reasons and the number of articles excluded in the first step of reading only the titles, both in the results section and figure 2.

- I am not so confident about the systematic review to the years 2018-2020. I think that should be an update in the literature (2018-2020), since there is only one paper (reference 11) in all review.

- The discussion section needs to be improved. More details are presented at specific comments.

Specific comments:

- Title: OK

- Abstract:

L.43: replace the word “citobrush” to “cytobrush”

L. 43: there is no need for the keyword “pregnancy”

- Introduction:

L. 56: “These quick, low cost diagnostic methods are commonly employed in clinical 56 practice, providing a subjective indication of uterine health”. I understand that the reference (10) is describing subclinical endometritis (SE) specifically and not uterine disorders in general (metritis, clinical endometrites, and or SE).

L. 61: I understand that “uterine lavage” should be replaced to the word “low volume uterine lavage”.

L. 62-64. In my point of view, it is not possible to make this simplistic consideration about the material collected in each technique.

L. 64. I disagree about the word “accurately”.

L. 66-67. “...and thus are better indicators of uterine health than the more commonly used methods of reproductive tract assessment”. I partially disagree with this statement, as I understand that the authors are talking about SE, and the techniques mentioned above (for example bacteriological culture) cannot be considered "better indicators".

L. 70: Please, consider to improve the figure 1 quality, mainly 1a, 1c, and 1d.

L. 79-80. “For endometrial biopsies (EB), the device is guided into each uterine horn and a small piece of tissue is excised [15, 21].”. Please, rewrite this sentence as it seems that the procedure performs a biopsy of each uterine horn in the same sample collection.

L. 98-99. I disagree with this statement “However, the impact of these procedures on the subsequent reproductive performance of cattle is unknown”, because there are some papers already published for this purpose, in addition to the papers that the authors identified in this study. For example see: 1) Bogado Pascottini, O., Hostens, M., Sys, P., Vercauteren, P., & Opsomer, G. (2017). Cytological endometritis at artificial insemination in dairy cows: Prevalence and effect on pregnancy outcome. Journal of Dairy Science, 100(1), 588–597. doi:10.3168/jds.2016-11529.

Materials and Methods

- The exclusion and inclusion criteria in the first step, where only the title is used to exclude references, must be clearer in the text.

- Results

L. 138: Please, consider to improve the figure 2 quality.

L. 139: Is this sentence right? “....1255 studies were excluded”.

L. 185: “...although significantly higher in primiparous than multiparous cows (54.3 vs. 38.5%, P< 0.05).” this sentence is just to sampled cows. Please, rewrite this sentence.

L. 195: Check this reference please “Etherington et al.”. I believe there was a mistake in this reference.

- Discussion

L. 218-220: I am not confident about uterine lavage (UL), please see primiparous results (Cheong et al. [18]), and the PR between interventions (EB; UL) at 60 days after breeding (Pugliesi et al., [25]).

L. 224:  delete the full stop.

L. 240: Check this reference please [72]. I believe there was a mistake here.

L. 240: I am not sure if this sentence is OK in relation to the methods, and the conclusion. Please, check! “Cheong et al.,[18] and Thome et al.,[27] performed UL and CB 4 hr after insemination without affecting pregnancy rates”.

L. 243-245: It seems that this sentence is out to the context of the paragraph, and also the main discussion.

L.256-259: Please, read the paper (Martins T, Pugliesi G, Sponchiado M, Gonella-Diaza AM, Ojeda-Rojas OA, Rodriguez FD, Ramos RS, Basso AC, Binelli M. Perturbations in the uterine luminal fluid composition are detrimental to pregnancy establishment in cattle. J Anim Sci Biotechnol. 2018 Sep 17;9:70. doi: 10.1186/s40104-018-0285-6. PMID: 30356865; PMCID: PMC6191683.).

L. 274-275: “They found that biopsied cows had higher pregnancy rate than non-biopsied cows (44.4 vs. 38.9, P=0.146, respectively)”. There is no significant difference between PR (P>0.05), also the authors said it in the paper.

L. 294-306: This part of the discussion regarding infertile women and the mechanical stimulus of inflammation, although interesting, does not add to the subject proposed in this review.

Author Response

(The authors gave the same response as above.)

Reviewer 3 Report

Dear authors,

After critiical reading and studying this submitted review which studied as a systemetic review the effect of uterine sampling procedures on subsequent pregnancy rate in cattle, I may conclude this mansucript is clean and well written and relative easy to follow and understand. One has to be aware of the aim of the study which only claims to review "systematically and summarize existing evidence related to the impact of these uterine sampling procedures on the likelihood of sampled cattle females becoming pregnant". No direct impact of the excistence of a type or severeness of uterine disease was taken into account.

General question: Can you comment on this (to my opinion important) limitation of the study?

Number of small remarks:

Line 249: to my knowledge the bovine mebryo arrives into the uterine horn after 4-5 days pc.

Line 267: add days post partum

Figure 1: photos have to be improved for better understanding

Figure 2: lay out is not appropriate, vague printed in current draft version of the manuscript.

Thanks!

Author Response

(The authors gave the same response as above.)
